**Data Availability Statement:** All relevant data are within the paper and its Supporting information files.

# Healthy lifestyle during pregnancy: Uncovering the role of online health information seeking experience

Rita Rezaee[1©], Ramin Ravangard[2©], Fahime Amani[3©], Arefeh Dehghani Tafti[4©], Nasrin Shokrpour[5©], Mohammad Amin Bahrami[2]*

1 Clinical Education Research Center, Shiraz University of Medical Sciences, Shiraz, Fars, Iran,
2 Healthcare Management Department, Shiraz University of Medical Sciences, Shiraz, Fars, Iran,
3 Healthcare Management Department, Shahid Sadoughi University of Medical Sciences, Yazd, Yazd, Iran,
4 Biostatistics Department, Kerman University of Medical Sciences, Kerman, Kerman, Iran, 5 English Department, Shiraz University of Medical Sciences, Shiraz, Fars, Iran

© These authors contributed equally to this work.
* aminbahrami1359@gmail.com

## Abstract

In the new era, many people seek their health-related information through the Internet due to the increasing access to this technology. Searching online health information can affect the health behavior. This study aimed to investigate the correlation between online health information-seeking behavior and a healthy lifestyle during pregnancy in a sample of Iranian pregnant women. This cross-sectional study was conducted among pregnant women admitted to health centers of Eghlid city, Fars province, Iran in 2019. A total of 193 women participated in the study. The required data were gathered using two validated questionnaires to measure the online health information-seeking behavior and the healthy lifestyle practices of the participants. The collected data were analyzed through descriptive statistics and Pearson correlation coefficient using SPSS version 22. Online health information experience and its subscales showed no statistical correlation with a healthy lifestyle. Age and education did not correlate with online health information-seeking behavior. Age had a statistical correlation with a healthy lifestyle, but education had the same correlation only with some subscales of a healthy lifestyle. The findings were surprising, suggesting that online health information-seeking behavior does not affect the lifestyle of pregnant women. These finding and probable explanations are discussed, but due to the limited literature on the subject, further studies are recommended to be conducted.

## Introduction

Pregnancy is an important period of women's life [1, 2]. Studies show that all women who are planning for pregnancy have at least one risk factor that can negatively affect pregnancy outcomes [3]. Many of these risk factors such as alcohol consumption, tobacco use, nutrition, and physical activity are lifestyle-related factors which can be modified [3]. The effects of lifestyle risk factors on pregnancy outcomes have been confirmed in many studies. A study on 385

**Funding:** The author(s) received no specific funding for this work.

**Competing interests:** The authors have declared that no competing interests exist.

pregnant women in Tabriz, Iran, (2010) has found that maternal nutritional status and physical activity during pregnancy are associated with birth weight [4]. Another meta-analysis has shown that maternal lifestyle factors such as pre-pregnancy Body Mass index (BMI), poor nutrition, and lack of physical activity are associated with poor maternal and neonatal outcomes [5]. It has been documented that positive healthy behaviors such as appropriate nutrition, adequate physical activity, vitamin intake, regular perinatal care, and health care utilization can have long-term positive effects on maternal and child health. In contrast, unhealthy behaviors can lead to a wide range of pregnancy complications and long-term adverse effects on maternal and child health such as preterm labor, mother's obesity and overweight, low birth weight, preeclampsia, hypertension, sudden abortion, and emergency cesarean section [1, 3, 5, 6]. Therefore, health decisions during pregnancy are important and can affect the life of the mother and baby [1]. Thus, promoting lifestyle behaviors that can reduce the negative consequences of pregnancy and protect individuals during this critical period of life has become an important aspect of public health research [2]. In this regard, due to the confirmed relationship between lifestyle modification during pregnancy and pregnancy outcomes, many studies have examined the correlation of intention to pregnancy and lifestyle improvement practices among women. However, the results are different and, in some cases, confusing [6]. A study on 430 women with planned pregnancy in Belgium reported that 83% of the participants had at least one change in their lifestyle behaviors in preparation for pregnancy. The modifications included behaviors such as smoking, alcohol consumption, caffeine consumption, nutritional status, weight control, and folic acid multivitamin supplements intake [7]. A similar study on 283 pregnant women in the Netherlands has showed that actively preparing for pregnancy is associated with choosing a healthier lifestyle by women during the preconception period [3].

Although both these studies and the others have shown that pregnancy intention to conceive correlates with lifestyle changes, few women who have planned for pregnancy change their health behaviors [3]. Also, some studies have shown that women's attempts to change their lifestyle for the purpose of preparing for pregnancy are affected by their socioeconomic status and medical history including education level, income status, perceived poverty, high-risk pregnancy, and abortion history [7]. Also, studies show that women who receive pre-pregnancy health information are more likely to modify their lifestyle behaviors during pregnancy than those not prepared for pregnancy [3]. Therefore, information-seeking behavior can be considered as a motivational facilitator for lifestyle improvements before and during pregnancy.

In recent years, the Internet has become a common source of health information for pregnant women [1, 8]. Various studies have shown that the Internet use is increasing among pregnant women due to some reasons such as availability and accessibility, ease of use, low cost, anonymity, ability to retrieve a large amount of information in a short time, and opportunity to find the support and live experiences of similar people [1, 2, 9, 10]. A study on 1347 pregnant women in India showed that 86% of the participants had used the Internet to retrieve pregnancy-related information [11]. A large multicenter study in Italy has found that almost all participants are pregnancy e-health users [9]. Also, a national survey in the United States has shown that about 75% of childbearing women search the pregnancy and infant information through the Internet [12]. Another study in Sweden has reported a similar prevalence of the Internet use among pregnant women. This study has reported the frequency of the Internet searching by pregnant women between one to 63 times a month [13]. Other studies from India [14], Turkey [15], Germany [16], US [17, 18], Saudi Arabia [19], China [20], and Australia [21] have also reported the same results. The results of three review articles have also confirmed these findings [1, 8, 12].

Studies in Iran also show that the use of the Internet to seek pregnancy-related information is increasing. A descriptive survey on 196 pregnant women in different months of pregnancy in Hamadan reported that 75% of the participants used search engines for seeking health information at least one or two times weekly [22]. Another qualitative study at 5 Gynecological hospitals in Tehran has shown that the Internet is one of the most important information sources for pregnant women [23]. Another study involving 188 pregnant women referred to health centers in Behshahr reported that approximately 70% of the participants had used online sources moderately to highly to find pregnancy information [24].

Pregnant women generally have different goals and motives for using the Internet as an information source, including overcoming barriers to accessing information, reassuring or supplementing traditional information sources, improving their understanding and knowledge promotion, better managing of the pregnancy risks, sharing information with others, social networking, and satisfying their informational needs [9, 11, 18, 19]. Also, they have different information needs. They seek various subjects from online sources [2, 25]. Studies show that some of the most important topics searched by pregnant women include fetal development, pregnancy sociology, general counseling about pregnancy time, nutrition during pregnancy, available social supports, pregnancy complications, diseases and risks during pregnancy, mental health, delivery methods, relationship with spouse and child, physical activity, pregnancy stages and changes, breastfeeding, maternity stories, maternal and child products sales, prenatal and personal health care, medications and supplements, tobacco use, and lifestyle [1, 2, 8, 11, 14, 15, 17, 19, 20, 23].

In summary, the results of studies regarding the most common topics searched by pregnant women indicate that although these topics change during pregnancy, lifestyle-related ones are among the most commonly searched topics. In this regard, a qualitative study in China reported that support for lifestyle modifications during pregnancy was one of the most important benefits of using the Internet from the perspective of pregnant women [20]. Another qualitative study in Iran has also shown that promoting a healthy lifestyle is one of the main reasons for Iranian women to seek online health information [26]. However, the overall effect of searching online pregnancy-related information on the actual changes in lifestyle during pregnancy has remained controversial. Although some studies have reported that online seeking pregnancy-related information affects lifestyle changes, other studies have not confirmed this relationship. Therefore, this study aimed to investigate the effect of online health information-seeking experience on the women's lifestyle during pregnancy. As noted, pregnancy is a transformational period in the life of every woman. During this period, a woman's body changes, and many questions arise about her own health and that of the infant as well as the lifestyle during pregnancy [1, 3, 27]. In the pregnancy, women want to be assured they will have a healthy pregnancy and therefore seek a lot of information. According to the documents, in this period, women use various sources of information including websites and social networks that are among the most common information sources for pregnant women [28]. Reports show that women are increasingly using the Internet for searching pregnancy-related information as well as sharing their information and experiences. Looking at the accessibility of the Internet indicates that nowadays maternal health information is easily and quickly accessible at any time and place [1–4]. The use of the Internet as a source of health information, among all groups of population including pregnant women has increased significantly during the Covid-19 epidemic due to restrictions such as limited face-to-face access to health professionals [29–31]. However, using the Internet also has many challenges. Leaving aside the issue of access to the Internet in some parts of the world, issues such as mothers' ability to retrieve and evaluate online information, their confidence in online information in spite of the vast amount of misinformation being disseminated through the Internet, and the

understandability of online information for mothers are the challenges in the effectiveness of this technology in promoting maternal health [3–7]. Therefore, health authorities and professionals should facilitate pregnant women's access to online health information which can help in improving maternal health [9, 13]. This, requires awareness of the various aspects of pregnant women's health information seeking behavior [9]. This study, provides such insights by examining the online health information seeking behavior of pregnant women in a religious and developing community. Therefore, it also has policy applications for health authorities.

## Materials and methods

### Participants, setting, and procedures

This study examined the correlation between online health information-seeking behavior and healthy lifestyle through a cross-sectional method during the first 6 months of 2019 among a sample of Iranian pregnant women. A total of 193 women participated in the study. Participants were recruited from 2 health centers of Eghlid city, Fars province using simple random sampling. All the participants were selected from the pregnant women who were in second and third trimester of pregnancy and had an active prenatal care record at the health center. Iran has an extensive primary healthcare network including the health centers at the first level. Health centers that have recently been renamed to Comprehensive Health Services Centers are affiliated to medical universities. These centers provide a comprehensive package of health services for the covered population including prenatal care for the pregnant mothers. Therefore, they are the most appropriate place to access the pregnant women. We recruited the study participants from these centers considering in case every pregnant woman had a pregnancy care record at the health center. The inclusion criteria were being in second and third trimesters of the pregnancy, having access to the Internet, and using it for pregnancy-related purposes at least once in the past month. The sample size was calculated for correlation studies using the results of a pilot study that was done prior to the main research. The required sample was calculated as 194, considering $\alpha = 0/05$, $\beta = 0/2$ and $r = 0/2$. Given that the 2 study settings covered approximately equal population, the sample size was allocated to the centers equally. Questionnaires were administered by a member of the research team who attended the health center; after explaining the objectives of the research and obtaining a written consent, he distributed the questionnaires among the study candidates. The administration of the questionnaires was done in the waiting room of the centers on the day of pregnancy visit. A total of 210 questionnaires were distributed, of which 193 were analyzed and 17 were removed from the analysis due to the incomplete answers. None of the participants had complicated pregnancy considering that the high-risk pregnant women were referred to the second level of care and did not remain in the first level for receiving prenatal care.

### Measures

The required data were gathered using two validated questionnaires:

1.  eHIQ (e-Health Information Questionnaire part 1 and 2): This scale was used to measure the online health information-seeking experience of the participants. The eHIQ, developed by Kelly et al. in 2015 as a tool to facilitate the measurement of the potential consequences of using websites containing different types of material across a range of health conditions, is a 2-part instrument with 37 items [32]. eHIQ-Part 1 includes 11 items related to general views of using the Internet in relation to health. 11 items of eHIQ-Part 1 have been grouped into 2 sub-scales named Attitudes towards online health information (5 items) and Attitudes towards sharing health experiences online (6 items). eHIQ-Part 2 includes 26 items

related to the consequences of using a specific health-related online source. 26 items of eHIQ-Part 2 also have been grouped into 3 sub-scales, including Confidence and identification (9 items); Information and presentation (8 items), and Understanding and motivation (9 items). In our study, the participants were asked to respond to the 26 items of eHIQ-Part 2 regarding the online sources which they have sought during their pregnancy to find the required information. Also, the participants were asked to score all items of both parts on a 5-point scale ranging from 'never' to 'always' scored 1–5. We used a standard 'forward-backward' procedure to translate the eHIQ from English into Persian. To demonstrate the content validity, we used the content validity ratio to quantify the extent of the experts' agreement. The reliability of the questionnaire also was confirmed before the study using Cronbach's alpha.

2. LSQ (Lifestyle Questionnaire): The LSQ, which was developed by Lali et al. in 2012 to assess healthy lifestyle practices, consists of 70 items [33]. LSQ is a multidimensional scale consisting of ten factors including activities related to physical health, exercise and fitness, weight control and nutrition, illness prevention, psychological health, spiritual health, social health, drug and alcohol avoidance, accident prevention (safety behavior), and environmental health. The used LSQ is a Persian-originated tool which has been developed by Mohsen Lali, an Iranian psychologist, and his colleagues. They have confirmed its validity and reliability in their study.

### Analysis

After completing the questionnaires, the collected data were analyzed through descriptive statistics (including means and standard deviations) and Pearson correlation coefficient using SPSS version 22.

### Ethical considerations

All participants provided a written informed consent to participate in the study and were assured that their personal information would be kept confidential. The ethical written consent forms were obtained from the participants. All the participants read and signed the consent before filling the questionnaire and returned it to the researcher. The content of the consent forms and the procedure were all approved by the ethics committee. Also, all the study procedures were conducted in accordance with the ethical principles of the Declaration of Helsinki. Considering the ethical approval, the work was approved by the ethics committee of Shahid Sadoughi University of Medical Sciences, Yazd, Iran (Approval code: IR. SSU.SPH.REC.1399.022).

### Results and discussion

The mean age of the participants was 31.3±6.61 years. Also, 16.1% of them had a high school degree, 47.3% had a diploma or some university degrees, 34.4% a bachelor's degree, and 2.2% had a degree above the bachelor's degrees. Also, 86% of the participants had at least one previous pregnancy experience.

The descriptive results of the studied women's information-seeking behavior are presented in Table 1. As shown in this Table, understanding and motivation subscale that examines the extent of understandability of online health contents for the participants and the extent to which their online information-seeking experience encourage them to play a more active role in their health had the highest mean. On the other hand, the Confidence and Identification subscale that examines the extent to which online information-seeking experience makes the

**Table 1. Online health information seeking scores of the participants.**

| Item | Mean ± SD |
|---|---|
| **eHIQ-Part 1** | |
| Attitudes towards online health information | 3.23±0.88 |
| Attitudes towards sharing health experiences online | 3.27±0.91 |
| **eHIQ-Part 2** | |
| Confidence and Identification | 2.93±0.87 |
| Information and Presentation | 3.38±0.75 |
| Understanding and Motivation | 3.57±0.93 |
| **eHIQ (total)** | 3.29±0.74 |

sense of confidence for the participants to explain their health concerns and discuss them with others gives them confidence that they are able to manage their health and the value they give to the online information had the lowest mean score.

Also, the descriptive results of the women's healthy lifestyle are presented in Table 2. As shown in the Table, the participants had moderate to good scores regarding the healthy lifestyle subscales. Other findings regarding the healthy lifestyle showed that although age had a statistically significant correlation with a healthy lifestyle, education level had the same correlation with only three subscales, including social health, accident prevention behavior, and environmental health.

Additional tests showed that e-HIQ and its subscales had no statistically significant correlations with the participants' age and education (Table 3).

The same tests showed that healthy lifestyle and its 2 subscales including "accident prevention behavior" and "environmental health" had a statistically significant correlation with age, while it did not correlate with education (Table 4).

The correlation coefficients of online health information-seeking behavior and its subscales with a healthy lifestyle are presented in Table 5. Based on the findings presented in this Table, eHIQ and its subscales showed no statistically significant correlation with a healthy lifestyle. This finding is surprising and suggests that accessing the Internet and seeking online health information did not affect the studied women's lifestyle even in the pregnancy course.

**Table 2. Lifestyle scale and subscale scores of the participants.**

| Item | Mean ± SD |
|---|---|
| Physical Health | 3.78±0.55 |
| Exercises and fitness | 3.15±0.80 |
| Weight control and nutrition | 3.63±0.69 |
| Illness prevention | 4.32±0.57 |
| Psychological Health | 4.08±0.73 |
| Spiritual Health | 4.34±0.65 |
| Social Health | 4.20±0.68 |
| Drugs and alcohol avoidance | 4.56±0.84 |
| Accident prevention behavior | 4.28±0.64 |
| Environmental Health | 4.25±0.68 |
| **Healthy life style (total score)** | 4.06±0.44 |

**Table 3. Correlations of e-HIQ and its subscales with age and education.**

| e-HIQ-Part 1 and 2 | Age | | Education | |
|---|---|---|---|---|
| | r | Sig. | r | Sig. |
| **eHIQ-Part 1** | 0.03 | 0.76 | 0.15 | 0.17 |
| Attitudes towards online health information | 0.06 | 0.58 | 0.16 | 0.14 |
| Attitudes towards sharing health experiences online | 0.03 | 0.77 | 0.13 | 0.21 |
| **eHIQ-Part 2** | 0.04 | 0.74 | 0.10 | 0.36 |
| Confidence and Identification | 0.06 | 0.61 | 0.01 | 0.90 |
| Information and Presentation | 0.00 | 0.99 | 0.09 | 0.39 |
| Understanding and Motivation | 0.03 | 0.78 | 0/09 | 0.38 |
| **eHIQ-Part (total)** | 0.00 | 0.98 | 0.13 | 0.23 |

**Table 4. Correlations of LSQ and its subscales with age and education.**

| Life Style Subscales | Age | | Education | |
|---|---|---|---|---|
| | r | Sig. | r | Sig. |
| Physical Health | 0.01 | 0.91 | 0.03 | 0.75 |
| Exercises and fitness | 0.01 | 0.94 | 0.00 | 0.95 |
| Weight control and nutrition | 0.01 | 0.90 | 0.14 | 0.19 |
| Illness prevention | 0.03 | 0.75 | 0.07 | 0.50 |
| Psychological Health | 0.10 | 0.36 | 0.13 | 0.21 |
| Spiritual Health | 0.18 | 0.09 | 0.15 | 0.18 |
| Social Health | 0.18 | 0.10 | 0.07 | 0.53 |
| Drugs and alcohol avoidance | 0.14 | 0.18 | 0.15 | 0.15 |
| Accident prevention behavior | 0.32 | 0.00* | 0.16 | 0.13 |
| Environmental Health | 0.31 | 0.00* | 0.07 | 0.51 |
| **Healthy life style (total score)** | 0.25 | 0.05* | 0.03 | 0.76 |

*Sig. at P<0.05

**Table 5. Correlations of online health information-seeking subscales with healthy lifestyle.**

| | Healthy lifestyle | |
|---|---|---|
| | r | P value |
| Attitudes towards online health information | 0.04 | 0.754 |
| Attitudes towards sharing health experiences online | 0.05 | 0.647 |
| Confidence and Identification | 0.12 | 0.274 |
| Information and Presentation | 0.15 | 0.159 |
| Understanding and Motivation | 0.21 | 0.055 |
| **eHIQ (total)** | 0.14 | 0.212 |

# Discussion

Pregnancy is a potentially changing period in the women's life that exposes them to new worries and responsibilities [34]. Maternal health status can affect pregnancy outcomes as well as short-term and long-term infant health [34].

Healthy lifestyle including physical activity, healthy eating, and weight control is the main predictor of health status [4, 5]. During pregnancy, women have a strong motivation to choose a healthy lifestyle to have a healthy infant. Even, some believe that pregnancy can be an opportunity for women to return to healthy lifestyle [34]. Therefore, supporting pregnant women in modifying their lifestyle is among the priorities and responsibilities of health care providers. Typically, health systems at different levels provide parts of these supports in routine prenatal cares such as pregnancy consults. Pregnant women have traditionally received the advice and information needed to improve their lifestyle during pregnancy from health professionals [5–7, 9, 13]. However, accessing and obtaining health information have changed dramatically in recent years. Today, many people, including pregnant women, receive vast amounts of health information related to lifestyle from the Internet and social media. They also share a lot of information and experience with others through the Internet [11, 15]. This change creates opportunities and challenges. Taking advantages of the potential opportunities requires having a clear picture of its various aspects, including preferred and common sources of health information of pregnant women, their information needs, information retrieval methods, information validation process and criteria, level of trust in the information obtained and understanding them, and ultimately the practical use of information for health promotion purposes [19–24]. In this study, some of these aspects and their correlations with the healthy lifestyle were studied in a sample of Iranian pregnant women.

The findings showed that study participants had a moderately positive feeling to use the Internet as a source of pregnancy-related information in their health decisions and to share their experience online. There are many studies which have revealed that the majority of pregnant women sought their needed health information through the Internet, but the results of studies on the willingness of pregnant women to share their experiences online were widely dispersed [13, 15, 19, 22, 23]. It seems that factors such as lack of trust in Internet information as well as incomprehensibility of online information affect these views [35]. Therefore, health care providers should pay attention to improvement of the view and trust of pregnant women in online pregnancy-related information. Empowerment of pregnant women to search the reliable information, sufficient attention to their information needs, the production and dissemination the need-based contents, social marketing of the online health platforms, and the active involvement of health professionals can improve the pregnant women's perception and increase their confidence in online health information [34–36].

In this study, we found that the age and education level of the participants did not have any significant correlations with their views of online health information. Regarding this findings, some studies have reported that variables such as the perceived barriers, age, education, level of knowledge, occupational status, number of pregnancies, pregnancy care model, self-efficacy, gestational age, geography and ethnicity have significant relationships with the amount and type of information searched online [1, 11, 12, 14, 15, 20, 21, 37, 38]. However, some studies have not confirmed such relationships and have reported that women using online information do not have a specific profile [16].

Participants also believed that searching for online health information did not give them the confidence that they could manage their health and share their health concerns with others. On the other hand, they thought that seeking online pregnancy-related information did not prepare them for what might happen to their health. Also, they valued online information as moderate. Trusting the accuracy of online information and also believing that seeking online information can help to improve the health care are among the most important affecting factors of online health information seeking behavior and using online information to promote health [34–36]. Therefore, confidence for online health information has been studied in many studies, and some have shown that the Internet users are suspicious about online information,

while others have identified the Internet as a reliable source [9, 10, 12, 15, 22]. Empowering women to validate the retrieved online information, monitoring the online contents by regulatory bodies, and more active participation of health professionals in the production of online health information can improve public trust in such information [34–36]. The mean score of information and presentation subscale was also moderate in our study. This means that the respondents cannot use health websites and their contents easily. This is also one of the most challenging barriers to the effective use of the Internet as a source of health information, which has been reported in many studies [39, 40]. Online information should be presented in a way that can be used by target users [39]. Developing appropriate guidelines and presenting the pregnancy-related information according to them, taking into account the users' views in designing health websites, providing information in a more systematic way, and using images and visual aids appropriately can be applied [35, 37, 38]. Mean score of the last subscale, named understanding and motivation, was moderate. This means that pregnancy-related information provided online for participants has not been fully understandable for them and has not encouraged them enough to play an active role in their health management. Generally, readability, understandability, reliability, and action-ability are the main criteria for evaluating the usefulness of online health information. Including visual aids such as simple images and charts in the online information to make them more understandable, using a common language, defining and explaining the terms appropriately, categorizing the information with specific titles, presenting information in logical sequence, and providing practical summaries can help to promote the understandability of online health contents for pregnant women, which can, in turn, encourage them to apply such information in their health promotion [39, 40].

Furthermore, the results of the present study also showed that online health information-seeking behavior and its dimensions had nosignificant correlations with the participants' lifestyle. This finding was the main objective of this study. The ultimate advantage of producing and disseminating online health information is using them in health promoting actions by users. However, our findings showed that online health information seeking is not playing such role among the study participants. One reason for this finding can be the participants' low trust in online information. A study in the Netherland showed that women who had obtained pre-pregnancy health information were more likely to modify lifestyle behaviors than those who were not prepared for pregnancy [3]. Some other studies from different countries have reported the same findings; altogether confirming that pregnant women's access to online maternal health information could encourage them to choose a healthier life style, which in turn improves maternal health [9, 11, 16, 20, 26, 41]. Overall, the present study showed that online health information-seeking behavior does not have a positive relationship with lifestyle behaviors during pregnancy. There are many explanations for this question that why the Internet search does not predict the lifestyle of pregnant women. Although the Internet search is widely used to retrieve health information by pregnant women, it also has its challenges and disadvantages. Some of the most important challenges can be as follows:

- Unreliability due to the wide dissemination of health misinformation: The Internet technologies provide a wonderful opportunity to disseminate the accurate health information and facilitate the dissemination of health misinformation [42]. It is a growing concern that health information obtained from the Internet is not always reliable and up to date. Online health misinformation can be harmful, confusing, and uncomfortable. The existence of marketing incentives, conflict of interests and bias in the production and dissemination of information complicate this challenge [3, 16, 18, 43].

- Inability to judge and interpret online health information: Many Internet users do not have enough ability and skill to understand and interpret the retrieved online information. This

inability reduces the impact of access to the health information on health decisions/health outcomes and in some cases even imposes adverse effects [1, 8, 12, 23, 43].

- Lack of a legal framework: The lack of a legal framework for producing, publishing, and monitoring online contents is also a major concern [9].

- Confidentiality and information security: The development of digital technologies has raised concerns about the privacy of personal information. In legal frameworks designed to monitor online space, the privacy and confidentiality of personal information must specifically be addressed [20].

Given these challenges, it seems that developing effective strategies for the optimal use of online capacities for the improvement of pregnant women's health is essential. Such strategies can be formulated at three levels:

1. User Level: Increasing pregnant women's access to reliable online health information should be considered as a strategy. Empowering pregnant women through training health promotion, improving their internet skills and familiarizing them with search methods and specialized sources and websites are also helpful [22, 23].

2. Expert Level: Health professionals need to understand that many people today get their health information from online sources. Therefore, they should be prepared to support pregnant women in retrieving, interpreting and using online information [25]. They should encourage pregnant women to share information and discuss them [19].

3. Institutional Level: At this level, there is an urgent need to establish legal frameworks for regular monitoring and verification of the accuracy and validity of online health information [9, 42]. It is also helpful to strengthen the online culture through the use of social marketing and promotion of interventions [43]. Although Pemberton and Goldblatt (1998) pointed out that "in the digital age, the concept of knowledgeable person should be accepted and his web-surfing skill should be applied" [8], the Internet technologies can serve to promote health when online health information is accessible, reliable and understandable for all. Finally, further research is needed to fully understand the online health information-seeking behavior of pregnant women, including developing policies and reducing challenges and barriers.

This study had some limitations alongside its strengths and applications. First of all, a cross-sectional study was done; therefore, the results are subject to the limitations of cross-sectional studies; second, we did not have a control group in the study, while doing a study with a control group can provide a better picture; third, the data analyzed in this study are self-reported, and forth the study was done in a specific socio-economic and cultural context which limits the possibility of generalizing the findings to the other societies.

## Conclusion

In this study, we found that the pregnant women moderately used the pregnancy-related online information in their health decisions and shared their experiences with others; they had a moderate trust in online health information. Also, they moderately understood the online health contents and those contents did not encourage them enough to play an active role in their health improvements. Also, we found that the online health information seeking behavior and its subscales had nostatistical correlations with lifestyle modification among the studied participants. These findings have many implications alongside its research and theoretical

implications for the health authorities and professionals who are interested in using the potential of the Internet to improve the population health, especially pregnant women.

## Supporting information

**S1 Dataset. Data sheet of the study.**
(SAV)

## Acknowledgments

Authors acknowledge the participants for their participation and the staff of health centers where the study was done.

## Author Contributions

**Conceptualization:** Rita Rezaee, Ramin Ravangard, Fahime Amani, Arefeh Dehghani Tafti, Mohammad Amin Bahrami.

**Data curation:** Rita Rezaee, Fahime Amani, Mohammad Amin Bahrami.

**Formal analysis:** Rita Rezaee, Ramin Ravangard, Fahime Amani, Mohammad Amin Bahrami.

**Investigation:** Rita Rezaee.

**Methodology:** Rita Rezaee, Ramin Ravangard, Fahime Amani, Arefeh Dehghani Tafti, Mohammad Amin Bahrami.

**Project administration:** Rita Rezaee.

**Software:** Rita Rezaee, Arefeh Dehghani Tafti.

**Supervision:** Rita Rezaee, Nasrin Shokrpour, Mohammad Amin Bahrami.

**Validation:** Rita Rezaee.

**Writing – original draft:** Rita Rezaee, Ramin Ravangard, Fahime Amani, Arefeh Dehghani Tafti, Nasrin Shokrpour, Mohammad Amin Bahrami.

**Writing – review & editing:** Rita Rezaee, Ramin Ravangard, Fahime Amani, Arefeh Dehghani Tafti, Nasrin Shokrpour, Mohammad Amin Bahrami.

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
