## [Decision Letter · Decision Letter 0]

20 Nov 2021

PONE-D-21-21672Healthy Life Style During Pregnancy: Uncovering the Role of Online Health Information Seeking ExperiencePLOS ONE

Dear Dr. Bahrami,

Thank you for submitting your manuscript to PLOS ONE. After careful consideration, we feel that it has merit but does not fully meet PLOS ONE’s publication criteria as it currently stands. Therefore, we invite you to submit a revised version of the manuscript that addresses the points raised during the review process.

We look forward to receiving your revised manuscript.

Kind regards,

Manuela Bombana

Academic Editor

PLOS ONE

Journal Requirements:

a) Did participants provide their written or verbal informed consent to participate in this study?

6. We noticed you have some minor occurrence of overlapping text with the following previous publication(s), which needs to be addressed:

- https://www.researchsquare.com/article/rs-19224/v1

- https://medinform.jmir.org/2020/12/e23854/

The text that needs to be addressed involves the first paragraph of the results.

In your revision ensure you cite all your sources (including your own works), and quote or rephrase any duplicated text outside the methods section. Further consideration is dependent on these concerns being addressed.

Additional Editor Comments (if provided):

Dear authors,

Your manuscript has been reviewed by experts in the field. The paper should undergo major revisions.

Please find the referee reports below.

Please revise your manuscript according to the referees’ comments.

One of the referees has suggested that your manuscript should undergo extensive English revisions. Please address this issue during revision.

Please revise specifically the methodological part of the study with regard to the referees comments (ethics approval, sample size calculation, recriting and selection of participants etc.).

Please also revise intensively the introduction and discussion section with respect to the current state of the art and integrate and discuss recent studies in the field appropriately.

Please do not hesitate to contact us if you have any questions regarding the

revision of your manuscript or if you need more time. We look forward to

hearing from you soon.

Kind regards,

Manuela Bombana

Reviewers' comments:

Reviewer's Responses to Questions

**Comments to the Author**

1. Is the manuscript technically sound, and do the data support the conclusions?

Reviewer #1: Partly

Reviewer #2: Partly

2. Has the statistical analysis been performed appropriately and rigorously? 

Reviewer #1: Yes

Reviewer #2: I Don't Know

3. Have the authors made all data underlying the findings in their manuscript fully available?

Reviewer #1: Yes

Reviewer #2: Yes

4. Is the manuscript presented in an intelligible fashion and written in standard English?

Reviewer #1: No

Reviewer #2: Yes

5. Review Comments to the Author

Reviewer #1: The study “Healthy Life Style During Pregnancy: Uncovering the Role of Online Health Information Seeking Experience” is interesting. In this study the authors observed that Online health information-seeking does not affect women's lifestyle during pregnancy. Secondly, Online pregnancy-related information moderately encourages and motivates women to play an active role in their health improvement. Thirdly, Pregnant women's attitudes towards online health information and sharing experience online can be improved. Fourthly, Pregnant women did not have good confidence in online pregnancy-related information. Lastly, Online pregnancy-related information is not fully understandable and reliable for pregnant women.

However, the author must elaborate the contribution of this work in the introduction section. The motivation behind this work and how it will solve the real word problems. I would recommend the authors add some more explanations and also describe the innovation of this paper in the introduction section and a new section of conclusion must be added.

Moreover, attention should be given to the following highlighted points before resubmitting.

1. Attention should be given to the authors affiliation it must follow the journal standard.

2. The work presented in the discussion section is basically not discussion. This is the literature or may put in the introduction section. For reference “As to online information sharing, a study in Turkey reported that about half of its participants had shared the sought information with health professionals (15). A similar study in Hamedan, Iran, found that more than 75% of the pregnant women had shared their retrieved online information with others including health professionals, friends, and family members (22). Another study in Tehran, Iran, reported that all the studied participants tended to share information with others and assisted them in obtaining information (23). A similar study has also reported that 80.2% of its participants were eager to share the information they found (19). In contrast, Larsson's (2009) study showed that most women, who had searched online information, did not talk to others about the retrieved information (13). The ratio of pregnant women who share the retrieved pregnancy-related online information in a study in Saudi Arabia has been reported as 26.7%. This study has concluded that participants generally have a positive feeling about online information and think that the Internet information is easily understandable and interpretable (19).” The authors discussed their own work in the discussion section.

3. The paper may be proof read by a native speaker because it has many grammatical mistakes.

4. The contribution should be further enhanced in the introduction section.

5. The new section of conclusion may be added before the references.

All in all, I suggest accepting the paper after revision.

Reviewer #2: GENERAL. The manuscript is not paged for ease of assessment.

ABSTRACT: Under methods, line 3,is it valid or validated questionnaire? There is no need to include references in the abstract and abbreviations should 1st be written in full.

INTRODUCTION: The 2nd sentence line 1-2, paragraph 1 should be referenced. BMI should be written in full 1st also US. In paragraph 2,line 12,what do the authors mean by intention to pregnancy? The last sentence is too long and can be separated.

METHODS. This section should should be much more detailed. How many health facilities were used or the study, what were the facilities and what type of health facilities were they? What informed the choice of these particular facilities? How did the authors determine their sample size and select their study population? where specifically in the facilities were the interviews held, who administered the questionnaires and how was this done? Where there no inclusion or exclusion criteria? The eHIQ questionnaire by Kelly should be referenced .Which Medical University's ethical review committee approved the study? There is no information related to pregnancy with reference to the participants

RESULTS: It not clear where the statistically significant correlation between age with a healthy lifestyle and education with the 3 subscales is shown.

DISCUSSION: Did the study have any limitations?

6. PLOS authors have the option to publish the peer review history of their article (what does this mean?). If published, this will include your full peer review and any attached files.

Reviewer #1: No

Reviewer #2: No

---

## [Author Response · Author response to Decision Letter 0]

4 Jan 2022

Dear editor 

Hi and many thanks for your work on our submission, hope to be well. 

Respectfully, all the editors and reviewers' comments were considered carefully, the requested corrections were done as followings and highlighted in the submission file. Also, regarding the editor’s request for language editing, the manuscript was edited by Dr. Nasrin Shokrpour (Professor of teaching English at Shiraz University of Medical Sciences: https://isid.research.ac.ir/Nasrin_Shokrpour.) The English editions also, are track changed in the submission file. If any more revisions will be needed we are ready to do. 

Comment: Please ensure that your manuscript meets PLOS ONE's style requirements, including those for file naming. The PLOS ONE style templates can be found at https://journals.plos.org/plosone/s/file?id=wjVg/PLOSOne_formatting_sample_main_body.pdf and https://journals.plos.org/plosone/s/file?id=ba62/PLOSOne_formatting_sample_title_authors_affiliations.pdf

Response: It was done based on the journal style requirements. 

Comment: Please amend your current ethics statement to address the following concerns: a) Did participants provide their written or verbal informed consent to participate in this study? b) If consent was verbal, please explain i) why written consent was not obtained, ii) how you documented participant consent, and iii) whether the ethics committees/IRB approved this consent procedure.

Response: It was done; all of the requested descriptions were added to methods under the sub-title of “Ethical considerations”. 

Comment: In your Data Availability statement, you have not specified where the minimal data set underlying the results described in your manuscript can be found. PLOS defines a study's minimal data set as the underlying data used to reach the conclusions drawn in the manuscript and any additional data required to replicate the reported study findings in their entirety. All PLOS journals require that the minimal data set be made fully available. For more information about our data policy, please see http://journals.plos.org/plosone/s/data-availability. "Upon re-submitting your revised manuscript, please upload your study’s minimal underlying data set as either Supporting Information files or to a stable, public repository and include the relevant URLs, DOIs, or accession numbers within your revised cover letter. For a list of acceptable repositories, please see http://journals.plos.org/plosone/s/data-availability#loc-recommended-repositories. Any potentially identifying patient information must be fully anonymized. Important: If there are ethical or legal restrictions to sharing your data publicly, please explain these restrictions in detail. Please see our guidelines for more information on what we consider unacceptable restrictions to publicly sharing data: http://journals.plos.org/plosone/s/data-availability#loc-unacceptable-data-access-restrictions. Note that it is not acceptable for the authors to be the sole named individuals responsible for ensuring data access. We will update your Data Availability statement to reflect the information you provide in your cover letter.

Response: The data sheet of the study is submitted as supporting information. 

Comment: Your ethics statement should only appear in the Methods section of your manuscript. If your ethics statement is written in any section besides the Methods, please delete it from any other section. Please include your full ethics statement in the ‘Methods’ section of your manuscript file. In your statement, please include the full name of the IRB or ethics committee who approved or waived your study, as well as whether or not you obtained informed written or verbal consent. If consent was waived for your study, please include this information in your statement as well.

Response: It was done. All the requested explanations were added in methods under the sub-section of “ethical considerations”. 

Comment: We noticed you have some minor occurrence of overlapping text with the following previous publication(s), which needs to be addressed: https://www.researchsquare.com/article/rs-19224/v1, https://medinform.jmir.org/2020/12/e23854/. The text that needs to be addressed involves the first paragraph of the results. In your revision ensure you cite all your sources (including your own works), and quote or rephrase any duplicated text outside the methods section. Further consideration is dependent on these concerns being addressed.

Response: Thanks for your comment. It was considered carefully. The first paragraph of results was completely rewritten. Also, all the manuscript body was rechecked for this concern. For your more information I would like to explain that this manuscript is one of our series of works on the online information seeking behavior of different population groups. To date we have done more than 10 works on different groups including students; pregnant women; hospital patients, oral care consumers and other ones. Each of these works has defined as a unique research with its own design and unique ethics code. The first one was published in JMIR medical informatics which had been done on a sample of female high school students to examine the correlation of online information seeking behavior and the quality of life. Although, these works may overlap slightly; but they are independent studies on the different population groups. Thanks again. 

Editor's comments:

Comment: Dear authors, Your manuscript has been reviewed by experts in the field. The paper should undergo major revisions. Please find the referee reports below. Please revise your manuscript according to the referees’ comments. One of the referees has suggested that your manuscript should undergo extensive English revisions. Please address this issue during revision. Please revise specifically the methodological part of the study with regard to the referees comments (ethics approval, sample size calculation, recriting and selection of participants etc.). Please also revise intensively the introduction and discussion section with respect to the current state of the art and integrate and discuss recent studies in the field appropriately.

Response: All were done carefully. 

Reviewers' comments

Reviewer 1:

Comment: The study “Healthy Life Style During Pregnancy: Uncovering the Role of Online Health Information Seeking Experience” is interesting. In this study the authors observed that Online health information-seeking does not affect women's lifestyle during pregnancy. Secondly, Online pregnancy-related information moderately encourages and motivates women to play an active role in their health improvement. Thirdly, Pregnant women's attitudes towards online health information and sharing experience online can be improved. Fourthly, Pregnant women did not have good confidence in online pregnancy-related information. Lastly, Online pregnancy-related information is not fully understandable and reliable for pregnant women. However, the author must elaborate the contribution of this work in the introduction section. The motivation behind this work and how it will solve the real word problems. I would recommend the authors add some more explanations and also describe the innovation of this paper in the introduction section and a new section of conclusion must be added. Moreover, attention should be given to the following highlighted points before resubmitting.

Response: Thanks for the comment. The contribution of the work, its research and policy implications for health authorities and professionals and the motivation behind it were added to introduction (lines 137-166). Also, a new section of conclusion was added after the results and discussion. For these revisions, references were also updated. 

Comment: Attention should be given to the authors' affiliation it must follow the journal standard.

Response: It was done.

Comment: The work presented in the discussion section is basically not discussion. This is the literature or may put in the introduction section. For reference “As to online information sharing, a study in Turkey reported that about half of its participants had shared the sought information with health professionals (15). A similar study in Hamedan, Iran, found that more than 75% of the pregnant women had shared their retrieved online information with others including health professionals, friends, and family members (22). Another study in Tehran, Iran, reported that all the studied participants tended to share information with others and assisted them in obtaining information (23). A similar study has also reported that 80.2% of its participants were eager to share the information they found (19). In contrast, Larsson's (2009) study showed that most women, who had searched online information, did not talk to others about the retrieved information (13). The ratio of pregnant women who share the retrieved pregnancy-related online information in a study in Saudi Arabia has been reported as 26.7%. This study has concluded that participants generally have a positive feeling about online information and think that the Internet information is easily understandable and interpretable (19).” The authors discussed their own work in the discussion section.

Response: Thanks for the comment. The discussion section was revised and rewritten again. It was improved substantially. 

Comment: The paper may be proof read by a native speaker because it has many grammatical mistakes.

Response: the manuscript was edited by Dr. Nasrin Shokrpour (Professor of teaching English at Shiraz University of Medical Sciences: https://isid.research.ac.ir/Nasrin_Shokrpour.)

Comment: The contribution should be further enhanced in the introduction section.

Response: Thanks for the comment. The contribution of the work, its research and policy implications for health authorities and professionals and the motivation behind it were added to introduction (lines 137-166).

Comment: The new section of conclusion may be added before the references.

Response: A new section of conclusion was added before the references. 

Comment: All in all, I suggest accepting the paper after revision.

Response: Thanks for the kind decision, reviewers' comments contributed highly in the submission improvement. 

Reviewer 2:

Comment: GENERAL. The manuscript is not paged for ease of assessment.

Response: Sorry, it was done. 

Comment: ABSTRACT: Under methods, line 3,is it valid or validated questionnaire? There is no need to include references in the abstract and abbreviations should 1st be written in full.

Response: The corrections were made. 

Comment: INTRODUCTION: The 2nd sentence line 1-2, paragraph 1 should be referenced. BMI should be written in full 1st also US. In paragraph 2, line 12,what do the authors mean by intention to pregnancy? The last sentence is too long and can be separated.

Response: The reference of 2nd sentence, paragraph 1 was added. The words are written in full 1st. Regarding the term "intention to pregnancy, "most often, pregnancies are characterized as either “intended” or “unintended.” Intended pregnancies are those wanted at, or sooner than, the time they occurred. Unintended pregnancies include unwanted and mistimed pregnancies". (URL: https://www.ncbi.nlm.nih.gov/pmc/articles/PMC4734627/.). There are some studies which show that the mothers' intentions to pregnancy affect their health behaviors. The last sentence of introduction has been replaced in this version due to the revisions made to introduction section. 

Comment: METHODS. This section should be much more detailed. How many health facilities were used or the study, what were the facilities and what type of health facilities were they? What informed the choice of these particular facilities? How did the authors determine their sample size and select their study population? Where specifically in the facilities were the interviews held, who administered the questionnaires and how was this done? Where there no inclusion or exclusion criteria? The eHIQ questionnaire by Kelly should be referenced .Which Medical University's ethical review committee approved the study? There is no information related to pregnancy with reference to the participants. 

Response: Thanks for the comment. All were done. Study setting (including the number and type of health facilities that used to recruit the participants and the reason to choose them); sample size calculation and sampling method; the administration procedure of questionnaires; inclusion criteria and some information related to the pregnancy were explained in the "participant, setting and procedures" subsection of methods. The references of both questionnaires were added to" measures" subsection of methods and the references list were also updated. The name of ethical approval committee was added to "ethical consideration" subsection of methods. 

Comment; RESULTS: It not clear where the statistically significant correlation between age with a healthy lifestyle and education with the 3 subscales is shown.

Response: We added 2 new tables for the related results including:

Table 3. Correlations of e-HIQ and its subscales with age and education and 

Table 4. Correlations of LSQ and its subscales with age and education. 

Comment: DISCUSSION: Did the study have any limitations?

Response: Limitations were added to the end of discussion. 

With bet regards

---

## [Decision Letter · Decision Letter 1]

24 May 2022

PONE-D-21-21672R1

Healthy Life Style During Pregnancy: Uncovering the Role of Online Health Information Seeking Experience

PLOS ONE

Dear Dr. Bahrami,

Thank you for submitting your manuscript to PLOS ONE. After careful consideration, we feel that it has merit but does not fully meet PLOS ONE’s publication criteria as it currently stands. Therefore, we invite you to submit a revised version of the manuscript that addresses the points raised during the review process.

We look forward to receiving your revised manuscript.

Kind regards,

Vanessa Carels

Staff Editor

PLOS ONE

Reviewers' comments:

Reviewer's Responses to Questions

**Comments to the Author**

1. If the authors have adequately addressed your comments raised in a previous round of review and you feel that this manuscript is now acceptable for publication, you may indicate that here to bypass the “Comments to the Author” section, enter your conflict of interest statement in the “Confidential to Editor” section, and submit your "Accept" recommendation.

Reviewer #1: All comments have been addressed

Reviewer #2: (No Response)

2. Is the manuscript technically sound, and do the data support the conclusions?

Reviewer #1: (No Response)

Reviewer #2: Yes

3. Has the statistical analysis been performed appropriately and rigorously? 

Reviewer #1: (No Response)

Reviewer #2: I Don't Know

4. Have the authors made all data underlying the findings in their manuscript fully available?

Reviewer #1: (No Response)

Reviewer #2: Yes

5. Is the manuscript presented in an intelligible fashion and written in standard English?

Reviewer #1: (No Response)

Reviewer #2: Yes

6. Review Comments to the Author

Reviewer #1: (No Response)

Reviewer #2: INTRODUCTION: Line 76-78 should be recasted. The sentence in lines 148-150 does not seem complete. In lines 157 -159 , grammatical error should be corrected.

METHODS: The statement in lines 171-172 is not clear (who had referral; do the authors mean who were referred?). If so, this is at variance with the information in lines 176-177 which states that participants were recruited from 2 heath centres, hence the authors should clarify how and were the participants were recruited. If the sample size of the study was 194 (line 187), why did the authors distribute 210 questionnaires (line 193). It is not clear why the statement in lie 197 is in the methods section.

Line 199,what do the authors mean by valid questionnaires are they referring to validated questionnaires? Lines 201-204 and 219-220 should be referenced. Lines 238-239, where is the institution that granted the authors ethical approval for conduct of the study located?

RESULTS: Lines 243-244 is not clear, what's the difference between some University degrees and a bachelor degree?

DISCUSSION: The is no sub-heading titled discussion. It is not clear, whether the information after table 5 is a continuation of the results. Lines 386-398- In the the whole of this portion, the authors are just citing other studies and not discussing their findings and comparing them to others. This is just lengthening the discussion, hence this portion can be deleted.

7. PLOS authors have the option to publish the peer review history of their article (what does this mean?). If published, this will include your full peer review and any attached files.

Reviewer #1: No

Reviewer #2: No

---

## [Author Response · Author response to Decision Letter 1]

25 May 2022

Dear editor:

Hi and hope to be well; with thanks to the editors and reviewers of our submission, all the reviewers’ comments were carefully revised as followings. If, any more revisions will be needed we will ready to do. 

Reviewer #1: All comments have been addressed

Reviewer #2: 

INTRODUCTION: 

Comment: Line 76-78 should be recanted. 

Response: Thanks, it was done. 

Comment: The sentence in lines 148-150 does not seem complete.

Response: Thanks, It was done. 

Comment: In lines 157 -159, grammatical error should be corrected.

Response: Thanks, it was corrected. 

METHODS:

Comment: The statement in lines 171-172 is not clear (who had referral; do the authors mean who were referred?). If so, this is at variance with the information in lines 176-177 which states that participants were recruited from 2 heath centres, hence the authors should clarify how and were the participants were recruited.

Response: Thanks, It was done. 

Comment: If the sample size of the study was 194 (line 187), why did the authors distribute 210 questionnaires (line 193). 

Response: Given that response rate is not always 100% and all the distributed questionnaires are not returned completely, we distributed 210 questionnaires (more than the needed sample size) to achieve the target of 194 completed ones. 

Comment: It is not clear why the statement in lie 197 is in the methods section.

Response: Thanks, the sentence was moved to the results section. 

Comment: Line 199, what do the authors mean by valid questionnaires are they referring to validated questionnaires?

Response: Yes it is true, the correction was done. 

Comment: Lines 201-204 and 219-220 should be referenced.

Response: It was done (the references number 32 and 33)

Comment: Lines 238-239, where is the institution that granted the authors ethical approval for conduct of the study located?

Response: It was added. 

RESULTS: 

Comment: Lines 243-244 is not clear, what's the difference between some University degrees and a bachelor degree?

Response: We used some university degree for Associate degree, it was corrected. 

DISCUSSION: 

Comment: The is no sub-heading titled discussion. It is not clear, whether the information after table 5 is a continuation of the results.

Response: Thanks, it was corrected. 

Comment: Lines 386-398- In the whole of this portion, the authors are just citing other studies and not discussing their findings and comparing them to others. This is just lengthening the discussion, hence this portion can be deleted.

Response: It was done. 

With best regards; 

Correspondence

---

## [Decision Letter · Decision Letter 2]

12 Jul 2022

Healthy Life Style During Pregnancy: Uncovering the Role of Online Health Information Seeking Experience

PONE-D-21-21672R2

Dear Dr. Bahrami,

We’re pleased to inform you that your manuscript has been judged scientifically suitable for publication and will be formally accepted for publication once it meets all outstanding technical requirements.

Kind regards,

Aniekan Abasiattai

Guest Editor

PLOS ONE

Additional Editor Comments (optional):

The reviewer's have completed their reviews and the manuscript is acceptable for publication. The 2nd reviewer has noted some grammatical errors which should be corrected.

Reviewers' comments:

Reviewer's Responses to Questions

**Comments to the Author**

1. If the authors have adequately addressed your comments raised in a previous round of review and you feel that this manuscript is now acceptable for publication, you may indicate that here to bypass the “Comments to the Author” section, enter your conflict of interest statement in the “Confidential to Editor” section, and submit your "Accept" recommendation.

Reviewer #1: All comments have been addressed

Reviewer #2: (No Response)

2. Is the manuscript technically sound, and do the data support the conclusions?

Reviewer #1: Yes

Reviewer #2: Yes

3. Has the statistical analysis been performed appropriately and rigorously? 

Reviewer #1: Yes

Reviewer #2: I Don't Know

4. Have the authors made all data underlying the findings in their manuscript fully available?

Reviewer #1: Yes

Reviewer #2: Yes

5. Is the manuscript presented in an intelligible fashion and written in standard English?

Reviewer #1: Yes

Reviewer #2: No

6. Review Comments to the Author

Reviewer #1: (No Response)

Reviewer #2: GENERAL: There are still several grammatical errors that should be corrected before publication. Some are:

INTRODUCTION: Lines 74-75 should be ---Netherlands has showed that---. Lines 77-78 --intension to conceive correlates with---. Lines 78-80 should be ---change their lifestyle for the purpose of preparing for the pregnancy---. Line 140-150 should be -- their confidence in online information---.

METHODS: Line 173 should be --the inclusion criteria were being in---.

DISCUSSION: lINE 343 SHOULD BE --- the educational level of the participants did not have any----.347 health are should be health care. 349--CONFIDENCE FOR ON-LINE---. 353 should be ---more active participation of health---. 385 ----altogether confirming that---. Lie 388 should be --behavior does not have a---

7. PLOS authors have the option to publish the peer review history of their article (what does this mean?). If published, this will include your full peer review and any attached files.

Reviewer #1: No

Reviewer #2: No

---

## [Editor Report · Acceptance letter]

20 Jul 2022

PONE-D-21-21672R2 

Healthy Lifestyle during Pregnancy: Uncovering the Role of Online Health Information Seeking Experience 

Dear Dr. Bahrami:

I'm pleased to inform you that your manuscript has been deemed suitable for publication in PLOS ONE. Congratulations! Your manuscript is now with our production department. 

Kind regards, 

on behalf of

Dr. Aniekan Abasiattai 

Guest Editor

PLOS ONE